# α-parvin is required for epidermal morphogenesis, hair follicle development and basal keratinocyte polarity

**Johannes Altstätter**[1], **Michael W. Hess**[2], **Mercedes Costell**[3], **Eloi Montanez**[4]*

**1** Department of Molecular Medicine, Max-Planck Institute of Biochemistry, Martinsried, Germany, **2** Institute of Histology and Embryology, Innsbruck Medical University, Innsbruck, Austria, **3** Department of Biochemistry and Molecular Biology, University of Valencia, Burjassot, Spain, **4** Department of Physiological Sciences, Faculty of Medicine and Health Sciences, University of Barcelona and Bellvitge Biomedical Research Institute (IDIBELL), Barcelona, Spain

* emontanez@ub.edu

**Data Availability Statement:** All relevant data are within the manuscript and its Supporting Information files.

## Abstract

Epidermal morphogenesis and hair follicle (HF) development depend on the ability of keratinocytes to adhere to the basement membrane (BM) and migrate along the extracellular matrix. Integrins are cell-matrix receptors that control keratinocyte adhesion and migration, and are recognized as major regulators of epidermal homeostasis. How integrins regulate the behavior of keratinocytes during epidermal morphogenesis remains insufficiently understood. Here, we show that α-parvin (α-pv), a focal adhesion protein that couples integrins to actin cytoskeleton, is indispensable for epidermal morphogenesis and HF development. Inactivation of the murine *α-pv* gene in basal keratinocytes results in keratinocyte-BM detachment, epidermal thickening, ectopic keratinocyte proliferation and altered actin cytoskeleton polarization. *In vitro*, α-pv-null keratinocytes display reduced adhesion to BM matrix components, aberrant spreading and stress fibers formation, and impaired directed migration. Together, our data demonstrate that α-pv controls epidermal homeostasis by facilitating integrin-mediated adhesion and actin cytoskeleton organization in keratinocytes.

## Introduction

The epidermis of vertebrates is a stratified squamous epithelium, consisting of multiple layers of keratinocytes that are separated from the underlying dermis by the basement membrane (BM), a specialized extracellular matrix (ECM) rich in laminins and collagens [1]. The epidermis is continuously regenerated throughout the life of the organism, and its homeostasis is achieved through a balance between the proliferation of keratinocytes in the basal cell layer and the loss of differentiated cells from the external surface of the skin [2]. Once basal keratinocytes commit to a terminal differentiation program, they exit the cell cycle, detach from the BM and migrate upwards through the suprabasal layers. This cell transition is accompanied by the successive differentiation of basal keratinocytes into terminally differentiated corneocytes, which will eventually be shaded from the outer layer as dead squames [2]. Loss of basal keratinocytes is replenished by stem cells that reside in the basal layer of the epidermis, in the sebaceous glands and in the bulge of hair follicles (HFs) [3]. HFs and sebaceous glands are epidermal

**Funding:** This work was funded by the Max Planck Society and the DFG (MO-2562/1-2).

**Competing interests:** The authors have declared that no competing interests exist.

appendages that form the pilosebaceous unit, whose main function is the production of the hair shaft, which is enveloped by the inner-root-sheath (IRS) and the outer-root-sheath (ORS) of the HF [4]. In mice, HF morphogenesis is initiated during embryogenesis and is completed by postnatal day 14. Thereafter, HFs cycle between phases of apoptosis-dependent regression (catagen), quiescence (telogen), and growth (anagen) [4]. During HF growth, bulge-derived keratinocytes of the ORS migrate along the BM towards the distal HF bulb, where they come in close proximity with the dermal papilla, a specialized mesenchymal compartment enclosed by the hair bulb. In the hair bulb, ORS keratinocytes differentiate into highly proliferative hair matrix (HM) keratinocytes.

Integrin-mediated adhesion to ECM and signaling are essential for epidermal morphogenesis, HF development and cycling, and keratinocyte function. Basal keratinocytes mainly attach to the underling BM via α6β4 and α3β1 integrins [5]. The binding of BM with α3β1 results in the assembly of focal adhesions (FAs), whereas its binding to α6β4 results in the formation of hemidesmosomes [5]. In the skin, integrins regulate the formation and assembly of BM, as well as the proliferation and differentiation of keratinocytes [6–8]. The molecular mechanisms that contribute to these integrin-mediated processes are, however, only partially understood.

Parvins are a family of adaptor proteins that localize to FAs and facilitate the interaction of integrins with the actin cytoskeleton [9]. Parvins also interact with integrin-linked kinase (ILK), which binds to PINCH to form a ternary complex (IPP-complex) that is directly recruited to β1 and β3 integrin cytoplasmic domains, where it regulates integrin signaling [10]. α-parvin (α-pv) is expressed ubiquitously and is an essential regulator of actin-dependent processes, such as cell spreading and migration [11]. Constitutive deletion of *α-pv* gene in mice leads to embryonic lethality at midgestation associated to multiple cardiovascular defects [11]. Endothelial α-pv regulates cell-cell junction organization, apical-basal polarity and the assembly of the BM around blood vessels [12, 13]. The function of α-pv in basal keratinocytes *in vivo* and during epidermal morphogenesis has not been studied until now.

In the current study, we use keratinocyte-specific approaches to show that the murine *α-pv* gene is indispensable for epidermis and HF morphogenesis. Epidermal defects in α-pv deficient skin include epidermis-BM detachment, ectopic keratinocyte proliferation, impaired basal keratinocyte polarization, and delayed keratinocyte differentiation. Together, we conclude that α-pv is required for integrin-regulated processes in keratinocytes during epidermal and HF morphogenesis.

## Materials and methods

### Mutant mice

To delete α-pv in keratinocytes, K5-Cre transgenic mice [14] were bred into a background of α-pv$^{floxed/floxed}$ (α-pv$^{fl/fl}$) mice [12]. All experiments with mice were performed in accordance to German guidelines and regulations, and protocols were approved by the Committee on the Ethics of Animal Experiments of the Max Planck Society.

### Histology and immunohistochemistry

Histology and immunohistochemistry of skin sections was performed as previously described [15].

**Paraffin sections.** Back-skin was fixed for 24 hours at 4°C in 4% paraformaldehyde in PBS and embedded in paraffin. Individual 10-μm sections were mounted on adhesive glass slides coated with Poly-l-Lysine, dewaxed in xylene, and rehydrated in descending graded ethanol. Then, paraffin sections were incubated in blocking buffer (0.1% Triton X-100 and 3% BSA in PBS) for 1 hour at RT, followed by incubation with primary antibodies overnight at

4˚C. After washing 3× with 0.1% Triton X-100 in PBS for 15 minutes, secondary antibodies were applied for 1 hour at room temperature (RT). After washing 3× with PBS for 15 minutes, sections were embedded in Fluoromount.

**Frozen sections.** Unfixed back-skin was embedded in OCT (Shandon Cryomatrix, Thermo) and rapidly frozen. Individual 12-µm sections were mounted on glass slides. Then, cryosections were fixed with 4% paraformaldehyde, methanol or Zn-fixative (40 mM $ZnCl_2$, 3 mM calcium acetate monohydrate, 10 mM zinc trifluoroacetate hydrate, 100 mM Tris pH 6.8). Then, cryosections were incubated in blocking buffer for 1 hour at RT, followed by incubation with primary antibodies overnight at 4˚C. After washing 3× with 0.1% Triton X-100 in PBS for 15 minutes, secondary antibodies were applied for 1 hour at RT. After washing 3× with PBS for 15 minutes, sections were embedded in Fluoromount.

The following antibodies were used: rabbit anti-alpha-parvin (Dr. Reinhard Fässler [16]); mouse anti-paxillin (BD Biosciences); rat anti-E-cadherin (Zymed); rabbit anti-β-catenin (Sigma-Aldrich); rat anti-β4-integrin, FITC conjugated rat anti α6-integrin (PharMingen), Biotin-conjugated rat anti Mac-1 (PharMingen), PE-conjugated rat anti-Gr1 (PharMingen), rat anti-β1 integrin (Chemicon); rabbit antibody against LN332 (Dr. Monique Aumailley); rat anti-Ki67 (Dako); biotin-conjugated rabbit anti-phospho-Histone H3 (Upstate); rabbit anti-keratin 5 (Covance), rabbit anti-keratin 10 (Covance) and rabbit anti-loricrin (Covance). For secondary detection, species-specific Alexa Fluor-coupled secondary antibodies (Invitrogen) were used.

## Epidermal whole mounts

Whole mounts from tail-skin were prepared as previously described [15]. Small pieces of tail-skin were incubated in 5 mM EDTA in PBS at 37˚C for four hours. Subsequently, the epidermis was carefully peeled from the dermis and fixed in Zn-fixative at 4˚C overnight.

## Electron microscopy

Samples from back-skin were processed as described [17] by using immersion fixation with glutaraldehyde followed by $OsO_4$, and epoxy resin embedding.

## Isolation and culture of primary keratinocytes

Primary keratinocytes were isolated and cultured as previously described [15]. Keratinocyte growth medium was prepared from MEM medium (Sigma-Aldrich), complemented with 8% FCS, 45 µM $CaCl_2$, 5 mg/ml insulin (Sigma-Aldrich), 10 ng/ml EGF (Sigma-Aldrich), 10 mg/ml transferrin (Sigma-Aldrich), 10 mM phosphorylethanolamine (Sigma-Aldrich), 10 mM ethanolamine (Sigma-Aldrich), 0.36 mg/ml hydrocortisone (Calbiochem), 2 mM L-Glutamine (Invitrogen) and 1x Penicillin/Streptomycin (PAA). Keratinocytes were cultured on tissue-culture dishes coated with 10 µg/ml collagen-I (INAMED) and 10 µg/ml fibronectin (Merck). For immunostaining, cells were fixed in 3% paraformaldehyde, permeabilized with 0.1% Triton-X100 and incubated with blocking solution of 1 hour at RT.

## Adhesion assay

Primary keratinocytes ($1x10^5$ cells/well) were plated onto 96-well plates coated with poly-Lysine (Sigma-Aldrich), collagen-I (INAMED), fibronectin (Merck) or Laminin 332 (Dr. Monique Aumailley, University of Cologne, Germany). After 30 minutes incubation, non-adherent cells were removed through washing with PBS. The remaining adherent cells were lysed in substrate buffer (7.5 mM NPAG (Sigma-Aldrich), 0.1 M Na citrate pH 5, 0.5% Triton

X-100) overnight at 37˚C. The reaction was stopped by adding 50 mM Glycine pH 10.4, 5 mM EDTA, and the O.D. 405 was measured.

### *In vitro* wound healing assay

After 4 hours incubation in keratinocyte growth medium supplemented with 4 µg/ml Mitomycin C (Sigma-Aldrich), keratinocyte monolayers were gently scratched with the tip of a cell-scraper. Subsequently, images were captured every 10 min. for 24 hours at 37˚C and 5% $CO_2$ using a Zeiss Axiovert microscope. At least three independent experiments were performed and more than 20 individual cells were tracked for each experiment.

### Statistical analyses

Statistical analyses were performed using a two-tailed T-test. Values are presented as mean plus standard error of the mean. P-values lower than 0.05 (*), 0.01 (**) or 0.001 (***) were regarded as significant.

## Results

### Keratinocyte restricted deletion of α-pv leads to progressive hair loss

To gain insight into the functions of α-pv in keratinocytes and during epidermal morphogenesis, we intercrossed mice carrying loxP-flanked α-pv gene (α-pv$^{fl/fl}$) [12] with mice expressing the Cre recombinase under the control of K5 promoter (K5-Cre) [14]. Intercrosses between α-pv$^{fl/+;K5-Cre}$ males and α-pv$^{fl/fl}$ females yielded viable α-pv$^{fl/fl;K5-Cre}$ (referred to herein as α-pv$^{ΔK}$) mice at expected Mendelian ratio. Immunofluorescence analysis in skin samples from 2-week-old control (α-pv$^{fl/+;K5-Cre}$ and α-pv$^{fl/+}$) mice showed strong expression of α-pv in basal keratinocytes of interfollicular epidermis and ORS of the HFs (Fig 1A). Immunofluorescence analysis of skin samples from α-pv$^{ΔK}$ mice showed a strong reduction in the α-pv signal when compared to control mice (Fig 1A). Western blot analysis of lysates from freshly isolated keratinocyte from 1-week-old α-pv$^{ΔKC}$ mice showed downregulation of α-pv expression when compared with lysates from controls mice (Fig 1B). No β-pv expression was detected in keratinocyte lysates (Fig 1B).

At birth, α-pv$^{ΔK}$ mice were indistinguishable from control littermates. At 1 week of age, in contrast to control mice, α-pv$^{ΔK}$ mice displayed an irregular skin pigmentation with a patchy hair coat (Fig 1C). At 2 weeks of age, while control mice have developed a homogeneous hair coat, α-pv$^{ΔK}$ mice displayed sparse hair with partial alopecia that persisted for about 4 weeks (Fig 1C). Thereafter, α-pv$^{ΔK}$ mice gradually lost their hair, resulting in complete and persistent alopecia at around 8 weeks of age (Fig 1B). Together, these data showed that deletion of α-pv in basal keratinocytes results in irregular skin pigmentation, gradual hair loss and persistent alopecia.

### Loss of α-pv triggers epidermal thickening and detachment, and impairs HF development

Histological analysis of back skin sections from 3-day-old control and α-pv$^{ΔK}$ mice did not reveal any significant defects in α-pv$^{ΔK}$ mice (S1 Fig). At later stages, α-pv$^{ΔK}$ mice displayed progressive epidermal thickening and locally confined epidermal detachments at the dermal-epidermal junction (Fig 2A and 2B). The analysis also revealed that HF development was severely impaired in α-pv$^{ΔK}$ mice (Fig 2A, 2C and 2D). At 2 weeks of age, HFs in control mice had an average length of 1100 µm and reached the subcutaneous fat layer (Fig 2A). In contrast, HFs in α-pv$^{ΔK}$ mice had an average length of 270 µm and only 32% of them reached the

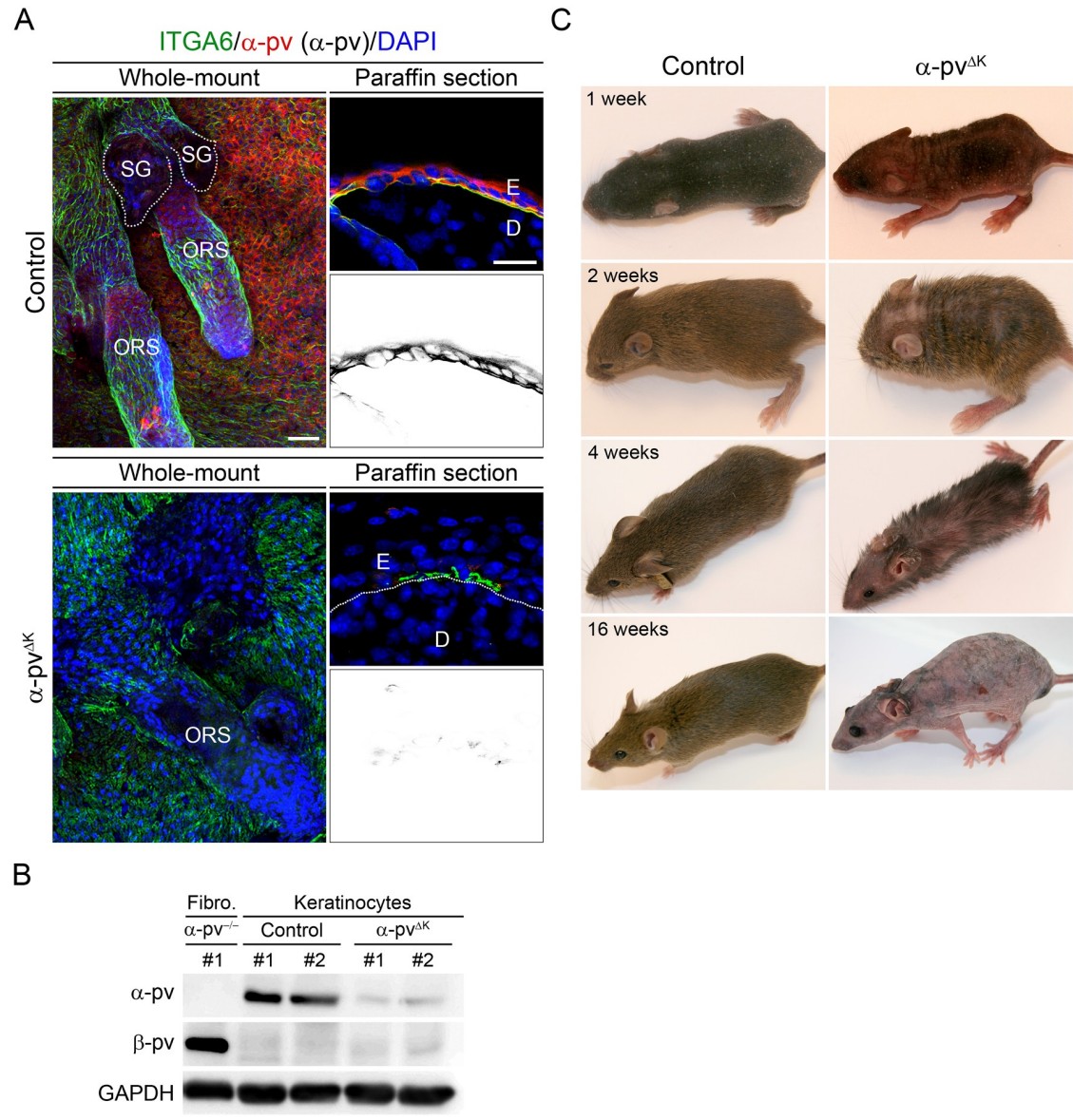

**Fig 1. Keratinocyte-restricted deletion of α-pv results in progressive and persistent hair loss.** (**A**) Tail-skin of 8-week-old (would-mount) and back-skin of 2-week-old (paraffin section) control and α-pv^ΔK mice stained for α-pv and α6 integrin. Scale bars are 40 μm and 20 μm, respectively. E: epidermis, D: dermis, ORS: outer-root-sheath and SG: sebaceous gland. Note the expression of α-pv in basal keratinocytes of interfollicular epidermis and ORS, and the strong reduction in the α-pv signal in α-pv^ΔK mice when compared to control mice. (**B**) Protein levels of α-pv and β-pv in keratinocyte lysates of control and α-pv^ΔK mice and in fibroblast (fibro) lysates isolated from α-pv^-/- mice [11]. (**C**) Gross morphology analysis of control and α-pv^ΔK mice at indicated time points.

subcutaneous compartment (Fig 2A and 2C). In addition, HF of α-pv^ΔK mice often appeared disorganized in structure, with distorted or absent hair shaft and dermal papilla, and enlarged sebaceous glands (Fig 2A). At 3 weeks of age, HFs of control and α-pv^ΔK mice were in the telogen phase, indicating that HF regression was not impaired in the absence of α-pv. However, while HF of control mice elongated during the following anagen phase and reached the subcutaneous compartment at 4 weeks, HFs of α-pv^ΔK mice did not undergo transition into the anagen phase and remained confined into the dermis (Fig 2A and 2B). Accumulations of melanin, which were most likely remnants of disintegrated HFs, were frequently detected in the dermis

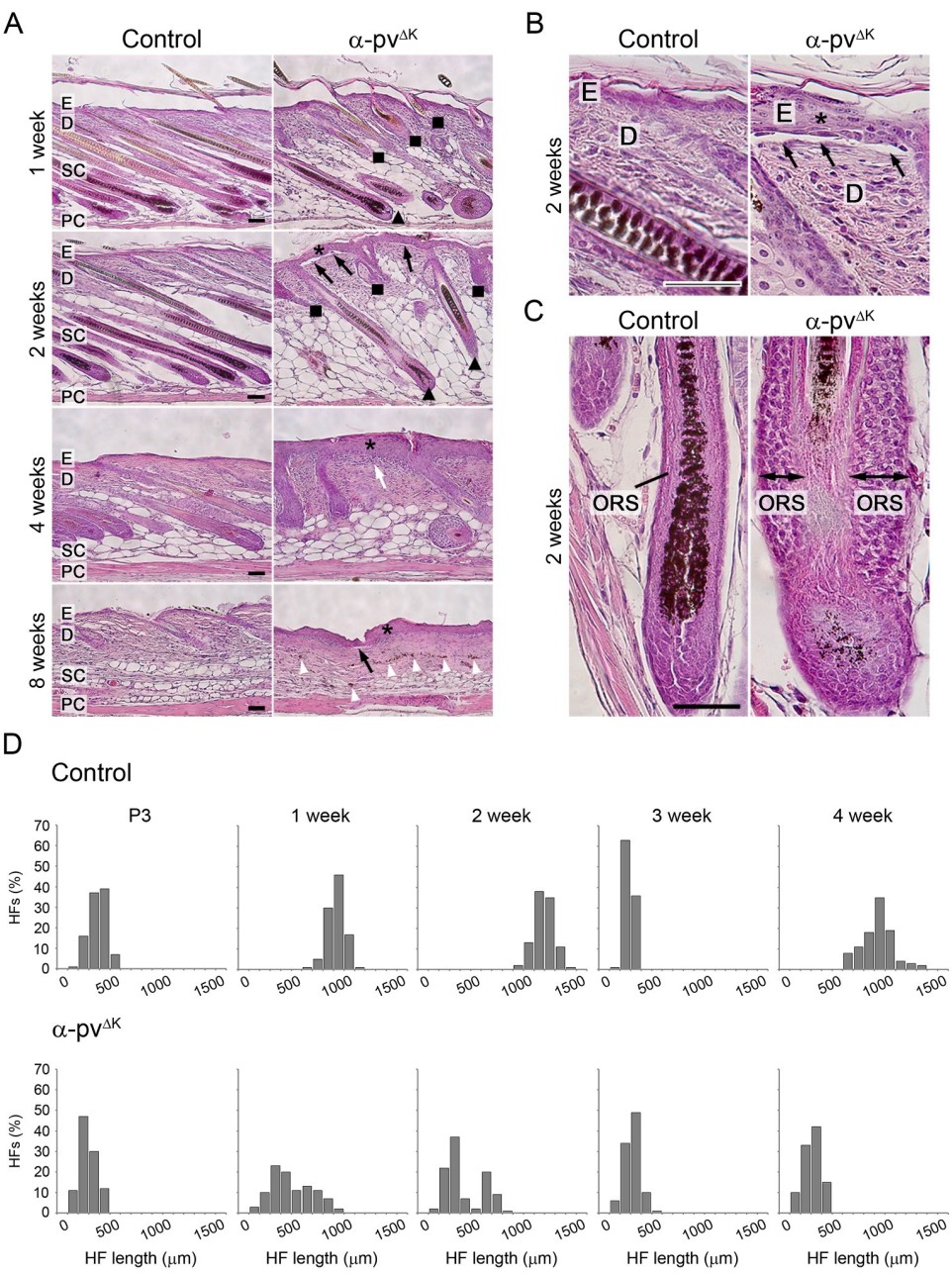

**Fig 2. Keratinocyte-restricted deletion of α-pv results in epidermis thickening and subepidermal blistering and impaired HF development.** (**A-C**) Hematoxylin-eosin staining of back skin section of control and α-pv^{ΔK} mice. Black arrows indicate areas of epidermal detachments at the DEJ, white arrow indicate subepidermal blister filled with repair tissue, white arrowheads indicate abnormal melanin-deposits. Triangle: fully developed HFs, square: short and prematurely growth arrested HFs and asterisk: epidermis hyperthickening. E: epidermis; D: dermis; SC: subcutis; PC: panniculus carnosum. Scale bar: 200 μm. (**D**) Histogram of HF length distributions at distinct stages of the HF cycle. At least 3 mice per genotype and time point were analyzed, and a minimum of 100 HFs per time point are presented in the histograms. Images were processed with the Image-J software.

and subcutis of 8-week-old α-pv$^{\Delta K}$ mice (arrowheads in Fig 2A). Together, these results indicate that α-pv is critical for epidermal homeostasis and HF development.

## Mice lacking α-pv show abnormal epidermal proliferation

To determine whether epidermal and ORS thickening observed in α-pv$^{\Delta K}$ mice were due to increased cell proliferation, we immunostained skin sections of 2-week-old control and α-pv$^{\Delta K}$ mice with antibodies against Ki67 and phospho-Histone-3, two cell proliferation markers. In control mice, proliferating keratinocytes were exclusively observed at the basal layer of the epidermis and around the dermal papilla in the HF (Fig 3A and 3B). In contrast, in the

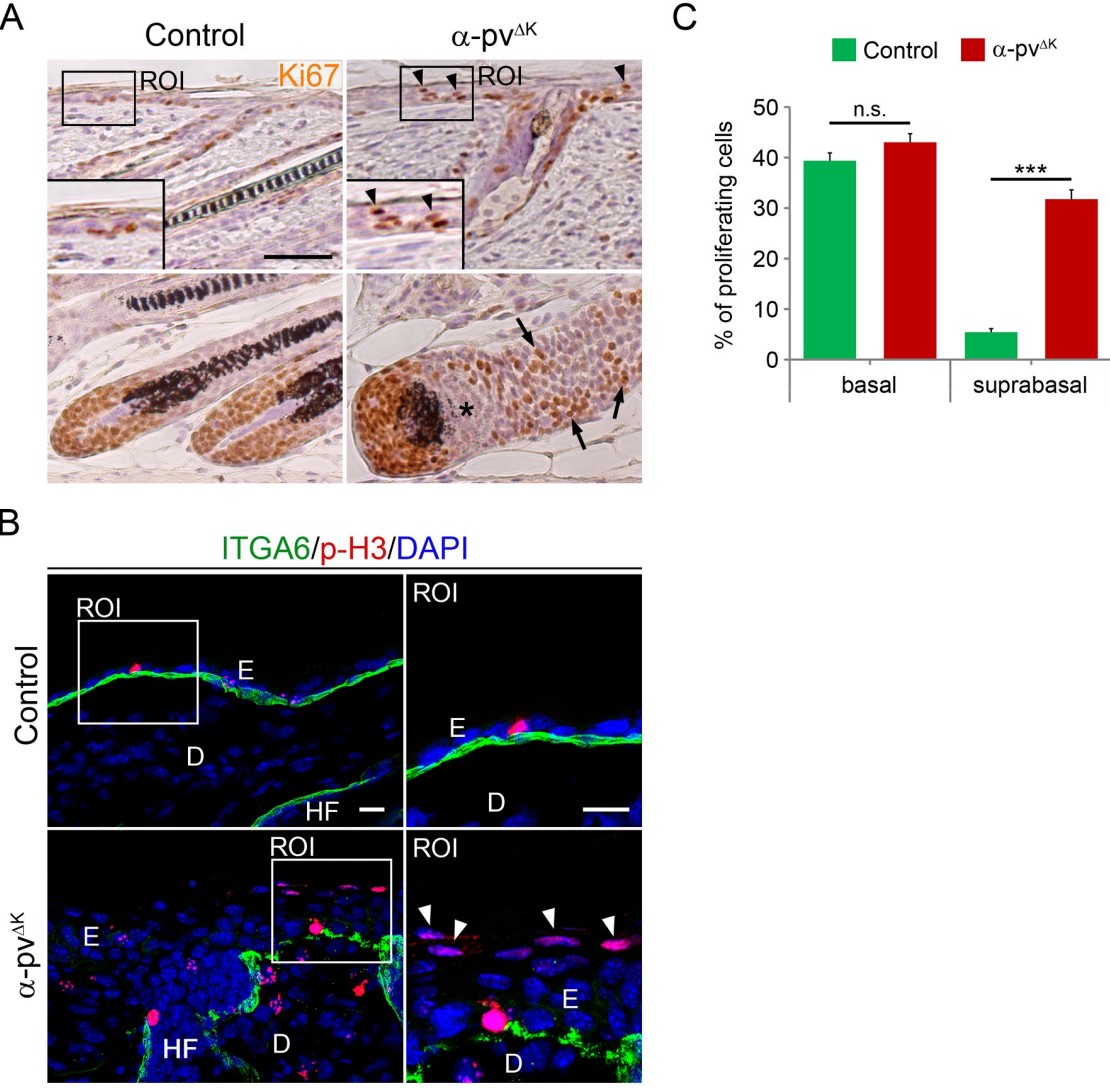

**Fig 3. Deletion of α-pv results in the accumulation of ectopically proliferating cells in suprabasal layers of the epidermis and in the ORS.** (**A**) Ki67 staining of control and α-pv$^{\Delta K}$ skin sections. Arrowheads indicate proliferating suprabasal cells and arrow point to proliferative cells in the ORS of α-pv-null HFs. Asterisks indicate a region in the α-pv$^{\Delta K}$ HF-bulb devoid of proliferative cells. Scale bar: 50 μm. (**B**) Double-fluorescent labeling for phospho-histone 3 and α6-integrin of control and α-pv$^{\Delta K}$ skin sections. Nuclei were visualized with DAPI. Arrowheads indicate proliferating suprabasal cells. Scale bar: 20 μm. E: epidermis, D: dermis and HF: hair follicles (**C**) Quantification of Ki67-positive cells in basal and suprabasal layers of control and α-pv$^{\Delta K}$ epidermis. Values represent means+SD.

epidermis of α-pv$^{\Delta K}$ mice, about 30% of proliferating keratinocytes were observed in suprabasal layers (Fig 3). The analysis also showed that the percentage of proliferating cells in the basal layer was not altered in α-pv$^{\Delta K}$ mice compared to control mice (Fig 3A and 3B). Moreover, in HFs of α-pv$^{\Delta K}$ mice, proliferative cells accumulated in the ORS and were partially absent from the hair bulb (Fig 3A). These data indicated that epidermal and ORS hyperthickening in α-pv$^{\Delta K}$ mice is due, at least in part, to ectopic proliferation of keratinocytes in suprabasal layers.

## Defects in integrin localization and BM organization in α-pv$^{\Delta K}$ mice

Integrin α6β4 is a central component of hemidesmosomes, specialized adhesion complexes that contribute to the binding of basal keratinocytes to the underlying BM and thus to the stable adhesion of the epidermis to the dermis [5]. The presence of subepidermal microblisters in α-pv$^{\Delta K}$ mice led us to analyze integrin localization in control and α-pv$^{\Delta K}$ mice. Co-immunostaining of skin sections from control mice for β4 integrin and the BM component laminin 332 (LN332) showed a thin and continuous BM, and a linear β4 integrin stain focally concentrated at the BM surface of the basal keratinocytes (Fig 4A; S2 Fig). In marked contrast, skin of α-pv$^{\Delta K}$ mice showed extensive areas with a discontinuous and disordered BM (Fig 4A; S2 Fig). Depletion of α-pv also severely altered the distribution of β4 integrin, which exhibited a discontinuous pattern and was undetectable in many of the regions with disordered BM (Fig 4A; S2 Fig). Moreover, while in control epidermis β1 integrin expression was restricted to basal keratinocytes, in the epidermis of α-pv$^{\Delta K}$ mice β1 integrin expressing cells were also found in suprabasal layers (Fig 4B; S2 Fig).

To examine whether loss of α-pv affects hemidesmosomes, we performed transmission electron microscopy analysis on back-skin sections from 2-week-old control and α-pv$^{\Delta K}$ mice. Skin of control mice displayed a distinct and continuous lamina densa immediately underneath basal keratinocytes, and numerous hemidesmosomes at the basal surface of basal keratinocytes (Fig 4C). In contrast, skin of α-pv$^{\Delta K}$ mice showed a disrupted BM and absence of hemidesmosomes in areas where basal keratinocytes had detached from the BM (Fig 4C). In addition, the analysis also revealed between the keratinocytes wider intercellular spaces with extensive interdigitating cell protrusions membrane in α-pv$^{\Delta K}$ mice compared to control mice (Fig 4C). Desmosomes were still present in α-pv-deficient epidermis, appeared morphologically normal, and mediated cell-cell contacts (Fig 4C). Taken together, these results indicate that loss of α-pv leads to hemidesmosome abnormalities and BM disruption.

Concomitant with the disruption of BM and the formation of epidermal blisters, granulocyte/macrophage infiltrates were observed in the skin of α-pv$^{\Delta K}$ mice in areas with discontinuous BM and around distorted HFs (Fig 4D). The absence of inflammatory signs in hyperplastic skin of 1-week-old α-pv$^{\Delta K}$ mice suggested that epidermal hyperplasia precedes inflammation rather than being a consequence of it (Fig 4D).

## α-pv is required for F-actin polarization

Detailed morphological analysis revealed that in basal keratinocytes from control mice, F-actin was predominantly accumulated at E-cadherin/β-catenin-positive adherens junctions to form the characteristic apical F-actin belt (Fig 5A). In contrast, BM-attached keratinocytes in α-pv$^{\Delta K}$ mice showed in addition to the apical F-actin belt a prominent F-actin accumulation at their basal surfaces, indicating that loss of α-pv affects F-actin polarization (Fig 5A). Associated to impaired F-actin localization and in contrast to control cells, BM-attached α-pv-deficient basal keratinocytes displayed unpolarized E-cadherin/β-catenin stain (Fig 5B and 5C). Together, these data indicated that α-pv is necessary for F-actin polarization in keratinocytes.

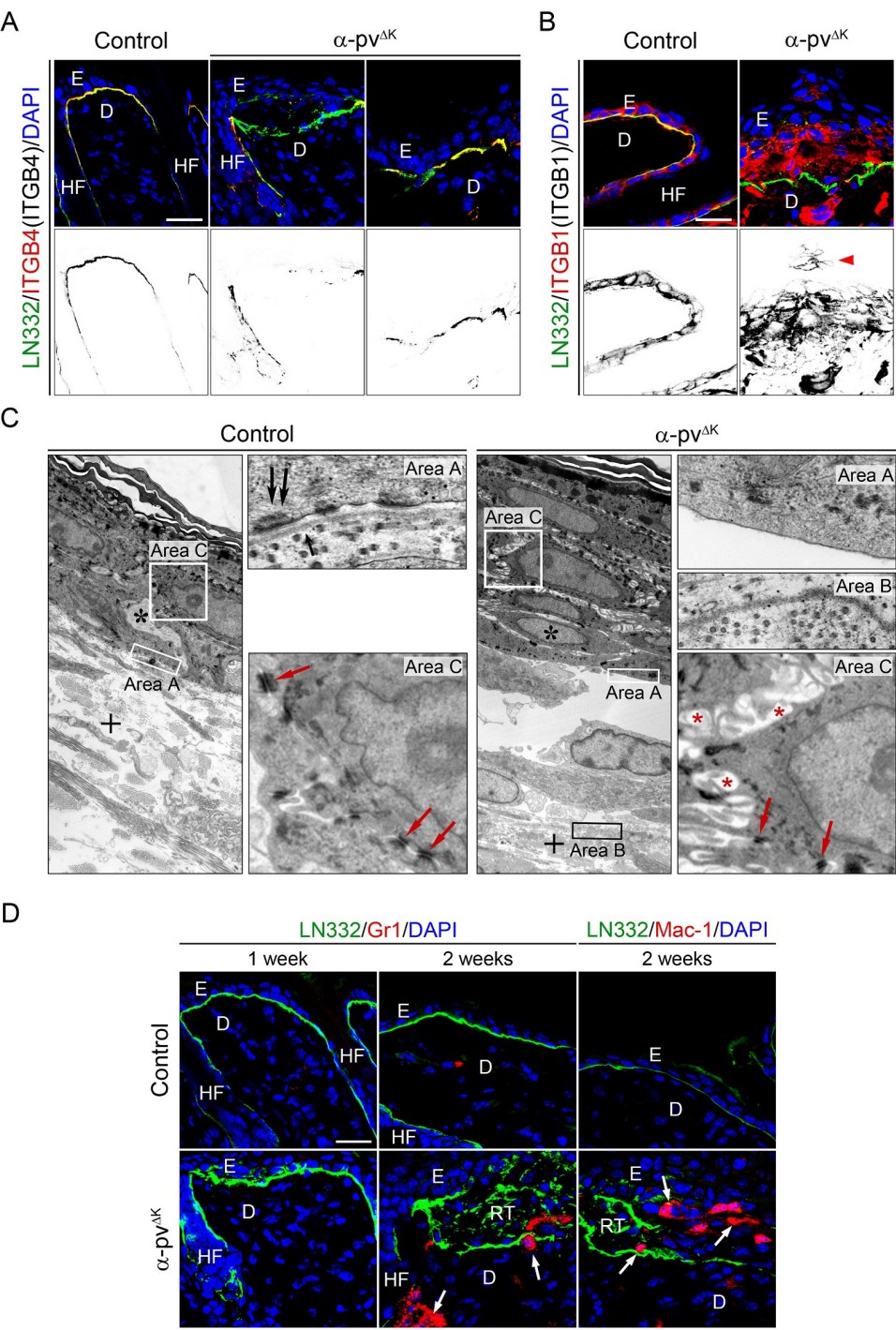

**Fig 4. Impaired integrin distribution and distorted BM organization in the skin of α-pv^ΔK mice.** Double-fluorescent labeling for (**A**) LN332 and β4-integrin, and (**B**) LN332 and β1-integrin of control and α-pv^ΔK skin sections. Nuclei were visualized with DAPI. Arrowhead indicates β1 integrin expressing suprabasal cells. Scale bar: 20 μm. (**C**) Ultrastructure of the skin of control and α-pv^ΔK mice. α-pv^ΔK mice are characterized by displacement of the basement membrane into the collagen-fibrils of the dermis (cross) and the widening of intercellular spaces (red asterisks), associated with the occurrence of microvilli-like cell protrusions. Double black arrows: desmosomes, single black arrow: lamina densa, red arrows: desmosomes and black asterisks: dermis. (**D**) Double-fluorescent labeling LN332 and Gr1, and LN332 and Mac-1 of control and α-pv^ΔK skin sections. Nuclei were visualized with DAPI. Arrows indicate Gr1-positive granulocytes and Mac1-positive macrophages in the dermis and adjacent to HFs in 2-week-old α-pv^ΔK mice. Scale bar: 20 μm. E: epidermis, D: dermis, HF: hair follicle and RT: repair tissue.

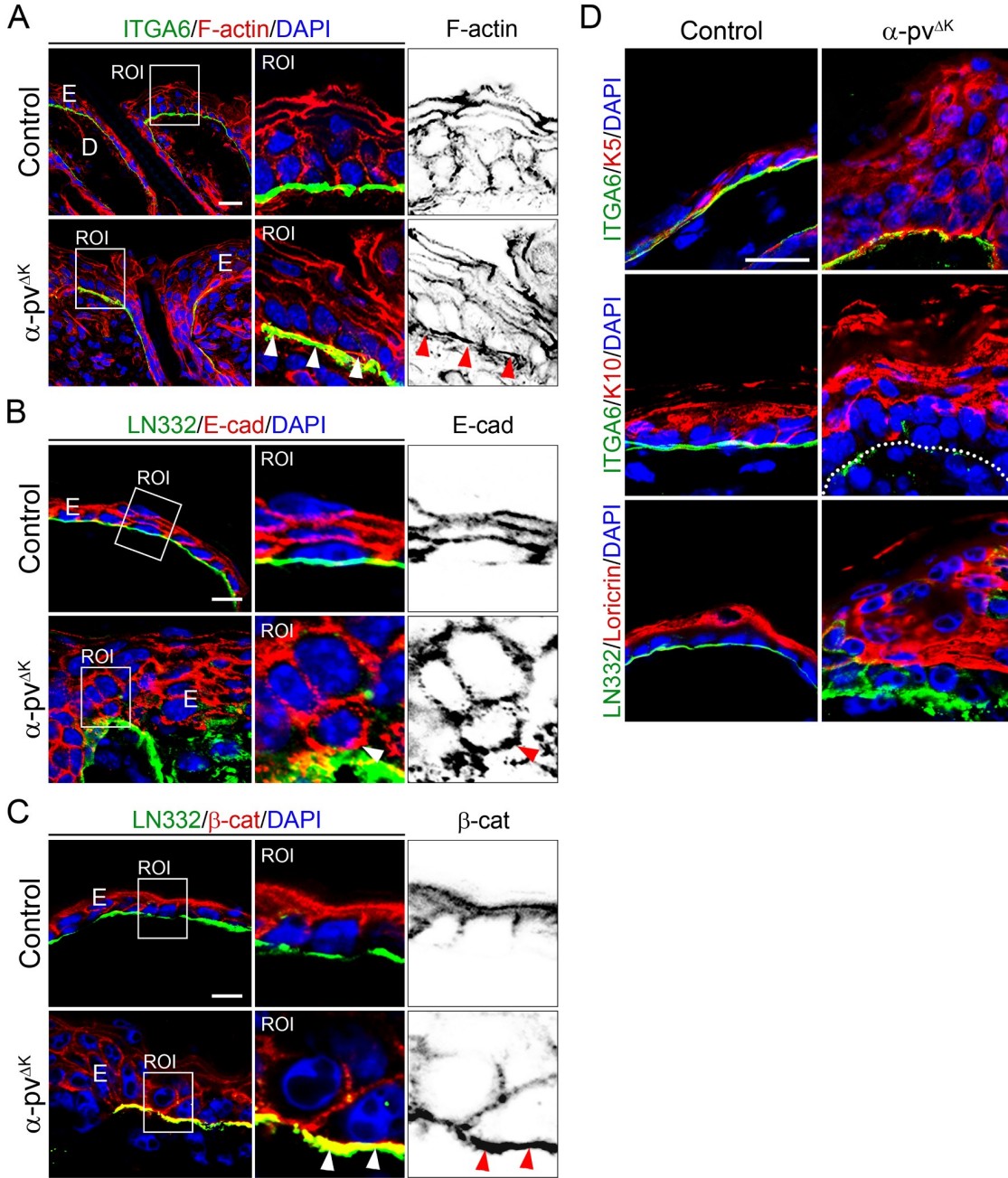

**Fig 5. Impaired keratinocyte polarity and differentiation in α-pv$^{\Delta K}$ mice.** Double-fluorescent labeling for (**A**) F-actin and α6-integrin, (**B**) E-cadherin and LN332, and (**C**) β-catenin and LN332 of control and α-pv$^{\Delta K}$ skin sections. Nuclei were visualized with DAPI. Arrowheads indicate basal stain of F-actin, E-cadherin and β-catenin in α-pv-null basal keratinocytes. Scale bar: 10, 20 and 20 μm respectively. (**D**) Double-fluorescent labeling for keratin-5 and α6-integrin, keratin-10 and α6-integrin, and loricrin and α6-integrin of control and α-pv-null skin sections. Nuclei were visualized with DAPI. Dotted line: basement membrane. Scale bar: 20 μm.

## Abnormal epidermal differentiation in α-pv$^{\Delta K}$ mice

Ectopic proliferation and expression of β1 and β4 integrins by suprabasal cells could be consequences of delayed differentiation and/or abnormal translocation of undifferentiated basal

keratinocytes to suprabasal layers. To investigate the effects of α-pv deficiency on epidermal differentiation, we examined the expression and localization of keratin-5 (a basal cell marker), keratin-10 (an early epidermal differentiation maker) and loricrin (a late epidermal differentiation maker) by immunofluorescence in skin sections of 2-weeks-old control and α-pv$^{ΔK}$ mice. While in control epidermis keratin-5 expression was restricted to the basal layer, in α-pv-deficient epidermis, keratin-5 expression was detected in basal keratinocytes as well as in suprabasal cells (Fig 5D; S3 Fig). Moreover, expression of keration-10, which is usually localized in the first suprabasal layer (spinous layer), was found in all suprabasal layers in α-pv-null epidermis (Fig 5D; S3 Fig). Finally, the expression of loricrin was no longer confined to the stratum granulosum but also present in nucleated cells below the stratum granulosum (Fig 5D; S3 Fig). Together these results suggested that deletion of α-pv impairs the terminal differentiation of keratinocyte.

## Deletion of α-pv compromises adhesion, spreading and migration of keratinocytes

To determine whether the poor adhesion of keratinocytes to the BM observed in α-pv$^{ΔK}$ mice was caused by the altered BM organization or by a defective integrin-mediated adhesion of the keratinocytes to the BM, we isolated keratinocytes from control and α-pv$^{ΔK}$ mice and tested their ability to adhere and spread on defined BM components. The analysis revealed that control keratinocytes readily adhered to fibronectin (FN), collagen type-I (Col-I) and LN332, while α-pv-null keratinocytes showed strongly and significant reduced adhesion to these substrates (Fig 6A). Integrin-independent adhesion to poly-L-Lysine was not affected in α-pv-null keratinocytes (Fig 6A). Moreover, most of the adhered α-pv-null cells remained rounded and those that were able to spread developed multiple unpolarized lamellipodia protrusions and failed to display stress fibers (Fig 6B). Loss of α-pv also let to pronounced reduction in paxillin-containing FAs (Fig 6C). To determine whether the lack of α-pv affects migration of keratinocytes, we monitored the polarized rate of cell migration during recovery of scratches introduced into primary keratinocyte monolayers. While control keratinocytes healed the wound within 24 hours, α-pv-null keratinocytes failed to efficiently migrate into the wound (Fig 6D). Single-cell tracking revealed two reasons for the impaired wound healing. First, while control keratinocytes migrated with an average speed of 0.56 ± 0.19 μm/min, α-pv-null keratinocytes were significantly slower and moved with an average speed of 0.34 ± 0.16 μm/min. Consequently, the accumulated distance migrated by α-pv-null cells (481.92 ± 227.85 μm) was significantly shorter than that of control cells (808.52 ± 275.12 μm). Second, α-pv-null keratinocytes were impaired in their directionality. The directionality index determined for control keratinocytes was 0.34 ± 0.14, whereas it was only 0.14 ± 0.09 for α-pv-null keratinocytes. Both defects together resulted in a significantly reduced euclidean migration distance of α-pv-null cells (62.59 ± 40.76 μm) compared to control (262.23 ± 107.13 μm) cells (Fig 6D and 6E).

## Discussion

Here, we show that α-pv is critical for epidermal and HF morphogenesis. Deletion of the *α-pv* gene in keratinocytes in mice has three main consequences: 1) keratinocyte-BM detachment resulting in subepidermal blisters and distorted BM organization, 2) impaired HF development and maintenance causing progressive and persistent alopecia, and 3) impaired epidermal differentiation manifested as epidermal hyperproliferation and thickening, and abnormal expression of epidermal differentiation makers.

   Adhesion of basal keratinocytes to the BM is mainly mediated by α6β4 and α3β1 integrins, which bind to LN-332, the main matrix component of the BM [5, 8]. Keratinocytes also

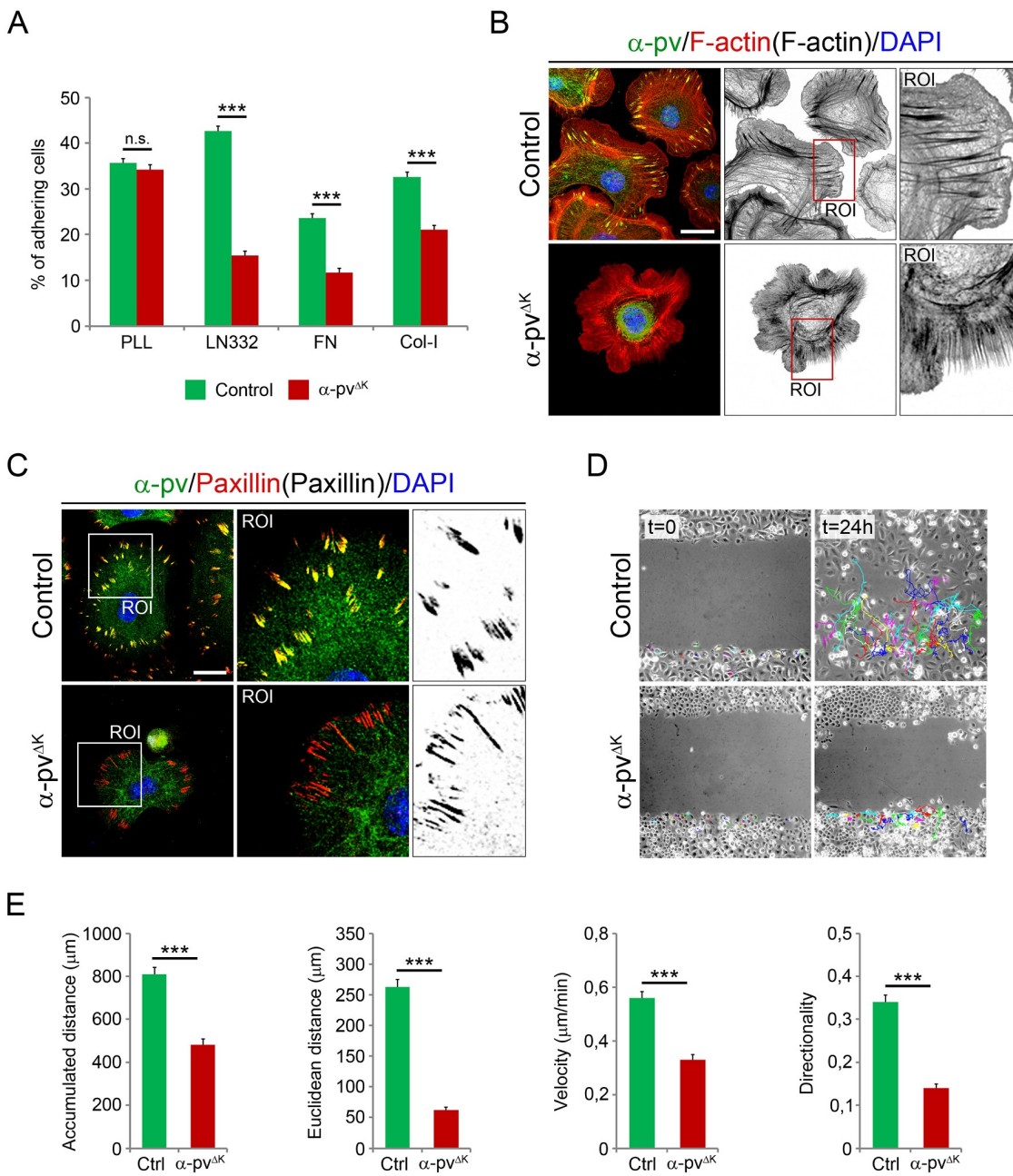

**Fig 6. Adhesion and migration of α-pv-deficient keratinocytes.** (**A**) Quantification of adhesion of control and α-pv-null keratinocytes to PLL, LN332, FN and Col-I. Values represent means of the percentage of adhering cells + SD. At least three independent adhesion assays were performed. (**B**) Double-fluorescent labeling for α-pv and F-actin of control and α-pv-null keratinocytes cultured on fibronectin. Nuclei were visualized with DAPI. (**C**) Double-fluorescent labeling for α-pv and paxillin of control and α-pv-null keratinocytes cultured on fibronectin. Nuclei were visualized with DAPI. (**D**) Scratch wounding of a confluent monolayer of control and α-pv-null keratinocytes. (**E**) Quantification of migration parameters as indicated. Values represent means +SD. At least three independent scratch assay experiments were performed and more than 20 individual cells were tracked in randomly chosen regions in each experiment.

express low levels of α2β1 and α5β1 integrins, which together with α3β1 are required for establish and maintain the integrity of the BM [6, 8, 18]. The interaction of LN-332 with β1 integrins results in the assembly of FAs, whereas its binding to α6β4 integrin results in the formation of hemidesmosomes [5]. The parvin-containing IPP complex associates with β1 and

β3 integrins, and thereby couples extracellular signals to a variety of intracellular processes such as signal transduction and cytoskeletal dynamics [9, 12, 13]. Here we show that α-pv$^{ΔK}$ mice displayed epidermal detachment and deterioration of BM integrity, indicating that α-pv regulates α3β1 integrin signaling *in vivo*. These results are in line with previous data showing that ILK and PINCH, the other members of the IPP-complex, are also key regulators of α3β1 integrin function during epidermal morphogenesis [19–21]. Although the IPP complex does not interact with β4 integrins, epidermal detachment in α-pv$^{ΔK}$ mice was associated to the dissociation of hemidesmosomes, suggesting that α-pv affects hemidesmosome stability indirectly. One possibility could be that BM defects caused by the absence of α-pv compromise hemidesmosome stability [22]. In addition, although α6β4 is the only integrin found in hemidesmosomes, other integrins can indirectly contribute to their assembly [8]. It has been shown that α3β1 integrin is involved in the assembly of hemidesmosomes *in vitro* [18, 22]. However, although BM integrity is severely compromised in α3-deficient mice, hemidesmosomes are intact and unaffected in these mice [23]. In contrast, conditional deletion of β1 integrins in keratinocytes results in significant reduction of hemidesmosomes, indicating that β1 integrins are necessary for the stability of the hemidesmosomes *in vivo*, and that α2β1 and α5β1 integrins can partially compensate the loss of α3β1 integrin in the epidermis [6, 8]. Together, these observations indicate that: 1) BM defects are not the primary cause of hemidesmosome abnormalities in α-pv$^{ΔK}$ mice, 2) α-pv not only regulates α3β1 integrin signaling, but also that of α2β1 and α5β1 integrins, and 3) α-pv is required to facilitate β1 integrin-dependent hemidesmosome stability. A requirement of α-pv for β1 integrin signaling was also evident in α-pv-deficient keratinocytes, which showed reduced adhesion to Col-I, FN and LN-332, diminished cell spreading, delayed formation of stress fibers and FAs, and impaired directed migration. Further experiments are now needed to understand how α-pv regulates the interplay between β1 integrins and α6β4 integrin during the assembly/disassembly of hemidesmosomes.

Epithelial cell polarity is characterized by the asymmetrical distribution of F-actin and cell junction proteins, such as E-cadherin and β-catenin [2]. In basal keratinocytes, F-actin localizes predominantly along cell-cell borders where it associates with E-cadherin/β-catenin complexes and stabilizes the adherens junctions [2]. Our results show that in α-pv-null basal keratinocytes F-actin was mainly distributed along the basal membrane, and localization of E-cadherin and β-catenin was no longer restricted to cell-cell borders, being also present at basal side. These defects were accompanied by the destabilization of the adherens junctions and the widening of the intercellular spaces. Interestingly, desmosomal structures were maintained in α-pv$^{ΔK}$ mice. Together, these results indicate that α-pv controls cell polarity of basal keratinocytes and that it is necessary to maintain the adhesive properties of adherens junctions. In line with this, it has been recently shown that α-pv also regulates cell-cell junction integrity and cell polarity in endothelial cells, a specialized type of epithelial cells [12, 13].

Loss of α-pv compromises HF morphogenesis and results in progressive loss of hair. While 35% of the HFs in α-pv$^{ΔK}$ mice completed their morphogenesis during the first 2 weeks after birth, the remaining 65% were severely distorted and prematurely growth-arrested. Asynchronous initiation and morphogenesis of distinct types of HFs together with the perinatal loss of α-pv can explain the development of both prematurely growth arrested and mature HFs in α-pv$^{ΔK}$ mice [24]. Although catagen and telogen appeared to proceed normal in HFs of α-pv$^{ΔK}$ mice, all HFs failed outgrowth during the following anagen phase. HF growth depends on proliferation and directional migration of stem cell-derived keratinocytes from the HF bulge region towards the HF bulb, where they differentiate into hair matrix keratinocytes. Directed migration is impaired in α-pv-deficient keratinocytes, suggesting that these migration defects account for the accumulation of proliferative keratinocytes in the ORS of mature α-pv-

deficient HFs, the impaired development of prematurely growth arrested α-pv-deficient HFs and the impaired HF outgrowth during anagen in the absence of α-pv.

During the course of terminal differentiation, keratinocytes undergo a series of changes in gene expression. Basal keratinocytes express keratin-5, keratin-14 and α6β4 integrin. As they detach from BM and migrate up to the spinous layer, the expression of these specific basal markers is stopped and the expression of keratin-1 and keratin-10 is activated. Finally, cells from the granular layer are filled with histidine- and cysteine-rich proteins, such as loricrin. In α-pv-deficient epidermis, cells in the suprabasal layers continued to express basal keratinocyte markers, including keratin-5 and α6β4 integrin. Moreover, nucleated cells below the granular layer expressed loricrin. This indicates that α-pv is necessary for the correct differentiation of keratinocytes. The α-pv$^{\Delta K}$ mice also displayed ectopic proliferation in suprabasal layers of the epidermis and inflammation. Inactivation of ILK or PINCH in keratinocytes in mice leads to epidermal defects resembling those in α-pv$^{\Delta K}$ mice [19–21], indicating that epidermal morphogenesis requires α-pv, ILK and PINCH-1, and therefore presumably also the formation of the IPP complex.

## Supporting information

**S1 Fig. Histological analysis of back skin sections from 3-day-old control and α-pv$^{\Delta K}$ mice.** Hematoxylin-eosin staining of back skin section of 3-days-old control and α-pv$^{\Delta K}$ mice did not reveal any significant differences between control and α-pv$^{\Delta K}$ mice. E: epidermis; D: dermis; SC: subcutis; PC: panniculus carnosum. Scale bar: 200 μm.
(TIF)

**S2 Fig. Impaired integrin distribution and distorted BM organization in the skin of α-pv$^{\Delta K}$ mice (overview images of Fig 4).** Double-fluorescent labeling for (**A**) LN332 and β4-integrin, and (**B**) LN332 and β1-integrin of control and α-pv$^{\Delta K}$ skin sections. Nuclei were visualized with DAPI. Scale bar: 20 μm. (**C**) Double-fluorescent labeling for LN332 and Gr1, and LN332 and Mac-1 of control and α-pv$^{\Delta K}$ skin sections. Nuclei were visualized with DAPI. Scale bar: 20 μm. E: epidermis, D: dermis, HF: hair follicle and RT: repair tissue.
(TIF)

**S3 Fig. Impaired keratinocyte differentiation in α-pv$^{\Delta K}$ mice (overview images of Fig 5D).** Double-fluorescent labeling for keratin-5 and α6-integrin, keratin-10 and α6-integrin, and loricrin and α6-integrin of control and α-pv-null skin sections. Nuclei were visualized with DAPI. Scale bar: 20 μm.
(TIF)

**S1 Raw images.**
(TIF)

## Acknowledgments

We thank Dr. Reinhard Fässler and Dr. Monique Aumailley who kindly provided us with antibodies.

## Author Contributions

**Conceptualization:** Michael W. Hess, Mercedes Costell, Eloi Montanez.

**Data curation:** Johannes Altstätter, Michael W. Hess, Eloi Montanez.

**Formal analysis:** Johannes Altstätter, Michael W. Hess, Eloi Montanez.

**Project administration:** Eloi Montanez.

**Supervision:** Eloi Montanez.

**Writing – original draft:** Eloi Montanez.

**Writing – review & editing:** Johannes Altstätter, Michael W. Hess, Mercedes Costell.

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
