## [Decision Letter · Decision Letter 0]

12 Feb 2020

PONE-D-20-01092

α-parvin is required for epidermal morphogenesis, hair follicle development and basal keratinocyte polarity

PLOS ONE

Dear Dr. Montanez,

Thank you for submitting your manuscript to PLOS ONE. After careful consideration, we feel that it has merit but does not fully meet PLOS ONE’s publication criteria as it currently stands. Therefore, we invite you to submit a revised version of the manuscript that addresses the points raised during the review process.

Your manuscript has been evaluated by two expert reviewers. Both reviewers find the manuscript novel and in general technically sound. There are some issues raised concerning microscopy images and data interpretation, that will have to be addressed to warrant publication. Lower magnification overview images should be included to facilitate interpretation of the high-magnification images, numbers should be provided for quantification/statistics of data, and conclusions drawn from the experiments in part require better support or may have to be adjusted (i.e. does inactivation of α-parvin really cause weakened keratinocyte-BM adhesion and rupture of the BM).

We would appreciate receiving your revised manuscript by Mar 28 2020 11:59PM. To enhance the reproducibility of your results, we recommend that if applicable you deposit your laboratory protocols in protocols.io, where a protocol can be assigned its own identifier (DOI) such that it can be cited independently in the future. For instructions see: http://journals.plos.org/plosone/s/submission-guidelines#loc-laboratory-protocols

We look forward to receiving your revised manuscript.

Kind regards,

Erik H. J. Danen

Academic Editor

PLOS ONE

Journal Requirements:

We note that one or more of the authors are employed by a commercial company: Roche Diagnostics GmbH.

Reviewers' comments:

Reviewer's Responses to Questions

**Comments to the Author**

1. Is the manuscript technically sound, and do the data support the conclusions?

Reviewer #1: Partly

Reviewer #2: Yes

2. Has the statistical analysis been performed appropriately and rigorously? 

Reviewer #1: I Don't Know

Reviewer #2: Yes

3. Have the authors made all data underlying the findings in their manuscript fully available?

Reviewer #1: Yes

Reviewer #2: Yes

4. Is the manuscript presented in an intelligible fashion and written in standard English?

Reviewer #1: Yes

Reviewer #2: Yes

5. Review Comments to the Author

Reviewer #1: This is the first report of an epidermis-specific inactivation of α-parvin. The claim in the title that α-parvin is required for epidermal morphogenesis, hair follicle development and keratinocyte polarity are justified based on the morphological images presented in the manuscript. In contrast, claims that inactivation of α-parvin results in weakened keratinocyte-BM adhesion and rupture of the BM are not supported by the data.

Major problems:

1. Most of the pictures are at high magnification. Consequently only a small portion of the section is shown while an overview at low magnification would allow a much better evaluation of skin defects. Observation at high magnification may have misled the authors. Indeed, there are blisters in Fig.2a, 2 weeks; and repair tissue filling the blister cavity in Fig.2a, 4 weeks, just below the asterisk; probably in Fig.2a, 8 weeks between the epidermis and the melanin deposits; and in Fig. 3b; in Fig. 4a, left photo of α-PVΔK between epidermis and the LN332-containing BM along the cavity floor; in Fig. 4b, photo of α-PVΔK between epidermis and the LN332-containing BM along the cavity floor (here the repair tissue is staining positive for beta 1 integrin as it is typical for repair tissue during wound healing); in Fig. 4d, the two pictures corresponding to α-PVΔK 2 weeks staining positive for LN332, Gr1 and Mac-1 (positive staining with Gr1 and Mac-1 is typical of the inflammatory activation during wound healing).

Several photos are centered on this repair tissue (b1 integrins, Mac-1, Gr1) and not on the epidermis as claimed by the authors.

2. Figure 1a,b aims to show parvin expression in skin. The origin of the rabbit antibody is not mentioned in Material and Methods. Very frequently rabbit antiserum contains antibodies against rabbit keratins. Staining of the paraffin section looks as a basal keratin staining, i.e. without polarization. The authors should show strictly parallel staining of control and KO skin side by side to be convincing. Where is parvin in the whole-mount? Are the keratinocytes used for the western blot cultivated or freshly isolated? And what about the fibro ? I guess it stands for fibroblasts which are not mentioned in the legend or in Material and Methods.

3. Figure 2 shows blisters at 2 weeks and not later. Is that correct? If yes what happen to the blisters at time point later than 2 weeks? Where are the blisters occurring? subepidermal or intraepidermal? and how much of the dermal-epidermal zone is affected. Overview should be provided. Explicit a, b and c separately in the legend. Details related to quantification of HFs are totally missing in the Legend or in Material and Methods (how many mice, how many sections per mice, which software, etc…).

4. Figure 3b, Figure 4a,b,d and Figure 5: For the same reasons as above, overview should be provided. As mentioned above, some pictures are centered to the repair tissue filling the blister cavity while others are centered on the epidermis outside the damaged areas where the dermal-epidermal junction appears to be intact. This is very confusing.

5. The blister level is not within the basement membrane plane since in Fig. 4 the electron micrographs of the knockout animals shown in c, there is a distinct basal lamina in area B. In addition for α-PVΔK skin, there is good positive staining of LN332 in the floor of the blister/filled blister cavity in Fig. 3b, Fig. 4a,b Fig. 4d5.

In conclusion, better illustrations should be provided and interpretation of the results needs a careful re-examination.

Reviewer #2: This is a nice manuscript describing the parvin knockout phenotype in skin. The analysis was interesting and well documented. The authors show that epidermal morphogenesis and hair follicle formation are critically dependent upon this key Integrin signalling component. I recommend publication.

6. PLOS authors have the option to publish the peer review history of their article (what does this mean?). If published, this will include your full peer review and any attached files.

Reviewer #1: No

Reviewer #2: No

---

## [Author Response · Author response to Decision Letter 0]

20 Feb 2020

2. Review Comments to the Author

We would like to thank all reviewers for their time, effort and valuable suggestions, which are greatly appreciated and have enabled us to improve the manuscript further.

Reviewer #1: This is the first report of an epidermis-specific inactivation of α-parvin. The claim in the title that α-parvin is required for epidermal morphogenesis, hair follicle development and keratinocyte polarity are justified based on the morphological images presented in the manuscript. In contrast, claims that inactivation of α-parvin results in weakened keratinocyte-BM adhesion and rupture of the BM are not supported by the data.

This is an important observation, thank you for the comment.

We have modified the abstract, the results and the discussion to avoid any misunderstandings. 

The revised abstract now reads “Inactivation of the murine α-pv gene in basal keratinocytes results in keratinocyte-BM detachment, epidermal thickening, ectopic keratinocyte proliferation and altered actin cytoskeleton polarization” and the revised discussion that now reads “Deletion of the α-pv gene in keratinocytes in mice has three main consequences: 1) keratinocyte-BM detachment resulting in subepidermal blisters and distorted BM organization…” 

Major problems:

1. Most of the pictures are at high magnification. Consequently only a small portion of the section is shown while an overview at low magnification would allow a much better evaluation of skin defects. Observation at high magnification may have misled the authors. Indeed, there are blisters in Fig.2a, 2 weeks; and repair tissue filling the blister cavity in Fig.2a, 4 weeks, just below the asterisk; probably in Fig.2a, 8 weeks between the epidermis and the melanin deposits; and in Fig. 3b; in Fig. 4a, left photo of α-PVΔK between epidermis and the LN332-containing BM along the cavity floor; in Fig. 4b, photo of α-PVΔK between epidermis and the LN332-containing BM along the cavity floor (here the repair tissue is staining positive for beta 1 integrin as it is typical for repair tissue during wound healing); in Fig. 4d, the two pictures corresponding to α-PVΔK 2 weeks staining positive for LN332, Gr1 and Mac-1 (positive staining with Gr1 and Mac-1 is typical of the inflammatory activation during wound healing).

Several photos are centered on this repair tissue (b1 integrins, Mac-1, Gr1) and not on the epidermis as claimed by the authors.

Thank you for this comment. The reviewer is correct and in many images it is possible to observe the blisters filled the repair tissue. To avoid any misunderstandings, we have depicted the repair tissue areas in the revised figure 2 and figure 4. Please also see point 3.

2. Figure 1a,b aims to show parvin expression in skin. The origin of the rabbit antibody is not mentioned in Material and Methods. 

We apologize for this shortcoming. The anti-alpha-parvin rabbit antibody was provided by Dr. Reinhard Fässler. We now provide this information in the revised material and methods section. We have also incorporated a reference into the material and methods in the revised manuscript (ref.17) 

Very frequently rabbit antiserum contains antibodies against rabbit keratins. Staining of the paraffin section looks as a basal keratin staining, i.e. without polarization. The authors should show strictly parallel staining of control and KO skin side by side to be convincing.

Thank you for this suggestion. We have performed immunostaining of skin samples from 2 weeks-old α-pvΔK mice with the anti-alpha-parvin antibody. The analysis revealed a strong reduction in the α-pv signal in α-pvΔK mice when compared to control mice. These results are displayed in revised figure 1A and mentioned in revised results sections.

Where is parvin in the whole-mount?

We apologize for not been clear in this point. As showed in figure 1A and mentioned in the results part, α-pv is expressed in basal keratinocytes of interfollicular epidermis and ORS of the HFs. To make this point clear, this is now also mentioned in the revised figure legend. 

Are the keratinocytes used for the western blot cultivated or freshly isolated? 

We apologize for this shortcoming. The keratinocytes used for the western blot were freshly isolated. This information is now mentioned in the revised results section.

And what about the fibro? I guess it stands for fibroblasts which are not mentioned in the legend or in Material and Methods.

The reviewer is correct, “fibro” stands for fibroblast. This information is mentioned in the revised figure legend.

3. Figure 2 shows blisters at 2 weeks and not later. Is that correct? If yes what happen to the blisters at time point later than 2 weeks? Where are the blisters occurring? subepidermal or intraepidermal? and how much of the dermal-epidermal zone is affected. 

We apologize for not been clear in this point. The subepidermal microblisters are also seen at 4 and 8 weeks. In the original figure 2A, the microblisters at 8 weeks were marked with a black arrow. In the revised figure 2A, the microblisters filled with repair tissue at 4 weeks are pointed with a arrow.

Overview should be provided. Explicit a, b and c separately in the legend. 

We believe that the overview images provided in figure 4A and the high magnification images provided in figures 4B and 4C adequately illustrate the skin defects in α-pvΔK mice.

Details related to quantification of HFs are totally missing in the Legend or in Material and Methods (how many mice, how many sections per mice, which software, etc…).

We apologize for this shortcoming. We provided all necessary information in revised legend of figure 2.

4. Figure 3b, Figure 4a,b,d and Figure 5: For the same reasons as above, overview should be provided. As mentioned above, some pictures are centered to the repair tissue filling the blister cavity while others are centered on the epidermis outside the damaged areas where the dermal-epidermal junction appears to be intact. This is very confusing.

Thank you for this suggestion. We provide overview images in revised figure 3B and new S2 and S3 figures. Moreover, to avoid any misinterpretations of the results, we have depicted the repair tissue areas in the revised figure 4.

5. The blister level is not within the basement membrane plane since in Fig. 4 the electron micrographs of the knockout animals shown in c, there is a distinct basal lamina in area B. In addition for α-PVΔK skin, there is good positive staining of LN332 in the floor of the blister/filled blister cavity in Fig. 3b, Fig. 4a,b Fig. 4d5.

Thank you for this comment. The skin of α-pvΔK mice showed extensive areas with a discontinuous and disordered BM, where LN332 was also detected in the dermal region below the BM plane. As we mention in the results section we focus our EM analysis in areas with distorted BM. 

In conclusion, better illustrations should be provided and interpretation of the results needs a careful re-examination.

Reviewer #2: This is a nice manuscript describing the parvin knockout phenotype in skin. The analysis was interesting and well documented. The authors show that epidermal morphogenesis and hair follicle formation are critically dependent upon this key Integrin signalling component. I recommend publication.

We thank the reviewer for recommending the publication of our manuscript without further modifications.

---

## [Decision Letter · Decision Letter 1]

28 Feb 2020

α-parvin is required for epidermal morphogenesis, hair follicle development and basal keratinocyte polarity

PONE-D-20-01092R1

Dear Dr. Montanez,

Your revised manuscript has been evaluated and we are pleased to inform you that all comments have been adequately addressed, your manuscript has been judged scientifically suitable for publication, and it will be formally accepted for publication once it complies with all outstanding technical requirements.

With kind regards,

Erik H. J. Danen

Academic Editor

PLOS ONE

Reviewers' comments:

Reviewer's Responses to Questions

**Comments to the Author**

1. If the authors have adequately addressed your comments raised in a previous round of review and you feel that this manuscript is now acceptable for publication, you may indicate that here to bypass the “Comments to the Author” section, enter your conflict of interest statement in the “Confidential to Editor” section, and submit your "Accept" recommendation.

Reviewer #1: All comments have been addressed

2. Is the manuscript technically sound, and do the data support the conclusions?

Reviewer #1: Yes

3. Has the statistical analysis been performed appropriately and rigorously? 

Reviewer #1: I Don't Know

4. Have the authors made all data underlying the findings in their manuscript fully available?

Reviewer #1: Yes

5. Is the manuscript presented in an intelligible fashion and written in standard English?

Reviewer #1: Yes

6. Review Comments to the Author

Reviewer #1: (No Response)

7. PLOS authors have the option to publish the peer review history of their article (what does this mean?). If published, this will include your full peer review and any attached files.

Reviewer #1: No

---

## [Editor Report · Acceptance letter]

3 Mar 2020

PONE-D-20-01092R1 

α-parvin is required for epidermal morphogenesis, hair follicle development and basal keratinocyte polarity 

Dear Dr. Montanez:

I am pleased to inform you that your manuscript has been deemed suitable for publication in PLOS ONE. Congratulations! Your manuscript is now with our production department. 

With kind regards,

on behalf of

Dr. Erik H. J. Danen 

Academic Editor

PLOS ONE